# Heat-Resistant Microporous Ag Die-Attach Structure for Wide Band-Gap Power Semiconductors

**DOI:** 10.3390/ma11122531

**Published:** 2018-12-12

**Authors:** Seungjun Noh, Hao Zhang, Katsuaki Suganuma

**Affiliations:** 1Department of Adaptive Machine Systems, Graduate School of Engineering, Osaka University, Osaka 567-0047, Japan; sjnoh@eco.sanken.osaka-u.ac.jp; 2The Institute of Scientific and Industrial Research, Osaka University, Osaka 567-0047, Japan; suganuma@sanken.osaka-u.ac.jp

**Keywords:** sinter joining, die attachment, wide band-gap semiconductor

## Abstract

In this work, efforts were made to prepare a thermostable die-attach structure which includes stable sintered microporous Ag and multi-layer surface metallization. Silicon carbide particles (SiC_p_) were added into the Ag sinter joining paste to improve the high-temperature reliability of the sintered Ag joints. The use of SiC_p_ in the bonding structures prevented the morphological evolution of the microporous structure and maintained a stable structure after high temperature storage (HTS) tests, which reduces the risk of void formation and metallization dewetting. In addition to the Ag paste, on the side of direct bonded copper (DBC) substrates, the thermal reliability of various surface metallizations such as Ni, Ti, and Pt were also evaluated by cross-section morphology and on-resistance tests. The results indicated that Ti and Pt diffusion barrier layers played a key role in preventing interfacial degradations between sintered Ag and Cu at high temperatures. At the same time, a Ni barrier layer showed a relatively weak barrier effect due to the generation of a thin Ni oxide layer at the interface with a Ag plating layer. The changes of on-resistance indicated that Pt metallization has relatively better electrical properties compared to that of Ti and Ni. Ag metallization, which lacks barrier capability, showed severe growth in an oxide layer between Ag and Cu, however, the on-resistance showed fewer changes.

## 1. Introduction 

To satisfy demands for high energy efficiency and reliability in extreme environments, wide band-gap (WBG) semiconductor materials including silicon carbide (SiC) and gallium nitride (GaN) have been adopted in power electronic systems [1,2]. It is understood that the power electronics can be significantly improved by using WBG semiconductors, due to the expectation of increasing power density, operating frequency, and break-down voltage. Moreover, the intrinsic characteristics of WBG semiconductors allow them to be used at high temperatures (>250 °C) due to their excellent electrical and physical properties [3,4,5]. The current interconnection materials are, for example, Pb-based solders, Sn-Ag, and Sn-Ag-Cu solders, of which the melting points are all below 300 °C. Si power electronics are not suitable for WBG power devices, due to their poor reliability in high temperatures [6,7,8]. Therefore, development of novel interconnection materials with high thermal and electrical conductivity and high melting temperatures is an urgent requirement for WBG semiconductors applied in severe thermal environments.

In order to address the increasing challenge of reliability issues for the high operating temperature of WBG power devices, corresponding thermal-resistant die-attachment solutions have been proposed. Silver (Ag) sinter joining paste has been recognized as the most competitive candidate due to its superior electrical performance, thermal conductivity, and high melting temperature [9,10,11]. We previously introduced low-temperature, low-pressure sintering joints by using Ag micro particles. Sakamoto et al. reported that micron-sized Ag pastes can be sintered at 300 °C with a low applied pressure of 0.4 MPa [12]. Suganuma et al. reported a low-pressure and low-temperature sintering joint using hybrid paste mixed with micron-sized Ag flakes and submicron-sized Ag particles [13], which has excellent stability under thermal shock at −40/300 °C in air. In addition, Zhang et al. reported a pressureless sinter joining paste that uses Ag micron particles and Ag_2_O nano particles, which results in a stable bonding structure at 180 °C [14]. Although the sintering process using Ag micron particles paste has various advantages, the microporous structure tends to change in morphology, for example, coarsening and interface “dewetting” when they are exposed to high temperatures [15,16,17]. 

In the previous study, Ag micron-sized flake paste containing SiC_p_ was used to inhibit coarsening of porous structures. The excellent stability of SiC_p_ in sintered Ag played a key role in preventing the morphological evolution of microporous structure and maintained its stable structure.

In this work, a thermostable bonding structure was fabricated with SiC dummy chips, SiC_p_-doped Ag paste, and direct bonded copper (DBC) substrates. This structure has been commonly used in Si power devices, such as insulated gate bipolar transistors (IGBT) [18]. The high temperature reliability of the four types of metallization structures were simultaneously evaluated by morphologies and on-resistance after high temperature storage (HTS) at 250 °C for 500 h and thermal cycling (TC) tests from −50 °C to 250 °C for up to 500 cycles. In addition, to clarify degradation after TC tests, thermal-mechanical simulation is then discussed in detail.

## 2. Materials and Methods

### 2.1. Preparation of SiC_p_-Doped Ag Sinter Joining Paste

Ag microflakes, with an average lateral diameter of 6 μm and a thickness of 80 nm (AgC-239, Fukuda Metal Foil and Powder, Kyoto, Japan), were used as starting material in this work. SiC_p_ with average diameter of 0.6 μm (HSC059, Superior Graphite, Chicago, IL, USA) was used as an additive. Table 1 presents the properties of Ag microflakes and SiC_p_, which are provided by the supplier. SiC_p_-doped Ag paste was prepared as follows. First, the Ag microflakes and SiC_p_ (2 wt.% of Ag flakes) were put in ethanol and further vibrated ultrasonically for 30 min to mix the components and prevent SiC_p_ aggregation. A uniform Ag flakes-SiC_p_ powder mixture was formed after evaporation of ethanol for 5 h at 50 °C in the oven. The desiccated powder was than mixed with ethylene glycol (EG) to obtain a 90 wt.% Ag paste.

### 2.2. Bonding Components and Procedures

A 3.1 × 4.4 × 0.35 mm^3^ SiC chip and 400 × 500 mm^2^ direct bonded copper (DBC) substrates were used to simulate practical joints in high temperature storage (HTS) tests and thermal cycling (TC) tests. Chips that were used in the present research were commercial SiC power semiconductors, such as the trench-gate metal-oxide-semiconductor field-effect transistor (TMOS) and Schottky diode (SBD). As indicated in Figure 1a, the bottom face of as-received SiC dies was metalized with a Ti/Ni/Au/Ag multi-layer metallization and the Ag outermost layer had a thickness of 300 nm. The as-received SiC dies were sputtered with a 1700 nm-thick Ag layer to adjust the Ag thickness to 2000 nm, however, the SiC dies used on Ni/Ag substrate were used as-received. The detailed chip metallizations are listed in Table 2. The thickness values of these metallization layers were confirmed by cross-sectional analysis of layers.

Ag paste was printed onto the DBC substrate by stencil printing using a metal mask. The thickness of the stencil used for paste printing was 100 μm. After mounting the SiC chip on the paste, the samples were placed on a hot plate at 250 °C under 0.4 MPa for 30 min, as shown in Figure 1b. Before the sintering process, the samples were preheated at 180 °C for 5 min to prevent unexpected void formation by rapid evaporation of EG.

The DBC substrate was composed of a silicon nitride ceramic plate (thickness: 0.32 mm) with thin Cu plates (thickness: 0.5 mm) bonded to both sides. Four types of DBC substrates with different metallization structures were simultaneously evaluated; one used electroless plating, and the others were sputtering. The electroless metallization structure was a Ni/Pd/Pt/Ag (5 μm/30 nm/300 nm/2 μm) multi-layers, while the sputtered metallization included Ag (2 μm), Ni/Ag (2 μm/2 μm) and Ti/Ag (500 nm/2 μm) multi-layer structures. The thermo-mechanical reliability of the sintered joints was evaluated by HTS tests at 250 °C up to 1000 h and by TC tests at −50 °C to 250 °C up to 500 cycles. The temperature profiles are presented in Figure 2. Under the operating conditions of WBG power devices, a junction temperature above 200 °C would be expected at the joint. In addition, extreme temperature variations could occur due to the transient heating up and cooling down. Thus, we performed HTS tests at 250 °C and TC tests at −50 °C to 250 °C.

### 2.3. Characterization of Sintered Joints

The die shear strength of the sintered joints was measured by shear test (Dage 4000, Nordan DAGE) at a shear rate of 100 µm s^−1^. The shear tool height from the DBC substrate was set as 50 μm. Five bonded specimens were evaluated to obtain mean value and standard deviation. The cross-section of the die-attach structure was prepared using an ion-milling polishing system (IM 4000, Hitachi, Tokyo, Japan), and its microstructure was observed by field-emission scanning electron microscopy (FE-SEM, SU8020, Hitachi, Tokyo, Japan) and energy-dispersive x-ray spectroscopy (EDS).

## 3. Results and Discussion

Figure 3 shows the microstructure evolution of pure Ag sintered joints at the chip sides during HST tests for 0 and 500 h. The initial sintered states have a microporous structure with a large number of Ag grains, as shown in Figure 3a–d. After HST tests for 500 h, the Ag grains grew into larger ones and the Ag porous structures became coarser (Figure 3e–h). The grain growth led to coarsening of porous structures by surface diffusion in order to reduce the surface area of the porous structures [19]. Meanwhile, Ag atoms’ diffusion also induced changes in the metallization on the SiC chips. The metallization layers became significantly thicker, while necking area deceased, indicating that thickening of chip metallization was caused by the expense of Ag atoms from the necking. It has been reported that the atoms’ diffusion from Ag porous network to chip metallization usually results in a weak-bonding layer, which leads to degradation of bonding structure [20]. Figure 4 shows the microstructure evolution of SiC_p_-doped Ag sintered joints at the chip sides during HST tests for 0 and 500 h. The microstructure of the as-sintered joints (Figure 4a–d) showed no obvious differences compared with the pure Ag joints. However, after HST tests for 500 h, the addition of SiC_p_ in the bonding structures inhibited coarsening in the porous Ag network and the thickening of chip metallization. The morphology remained similar to that of the as-sintered state, as shown in Figure 4e–h. This can be explained by the properties of SiC_p_, such as hardness and high temperature stability, which reduces the risk of void formation. It seems that a physical interaction between Ag and SiC_p_ maintained the original morphology of the porous Ag.

Figure 5 shows the microstructure evolution of pure Ag sintered joints at the substrates side in as-sintered state and after HTS for 500 h. Figure 5a–d show SEM images of the initial interface of Ag, Ni/Ag, Ti/Ag, and Ni/Pd/Pt/Ag metallization samples, respectively. The pure sintered Ag was well connected with Ag metallization on DBC substrates for the as-sintered structure, as shown in Figure 5a. After HTS tests for 500 h, an obvious Cu oxidation layer was formed between Ag metallization and Cu, as shown in Figure 5e. The source of oxygen was attributed to residual oxygen in the porous structure after the sintering process. During HTS tests, the oxygen penetrated through the grain boundaries in Ag metallization and formed Cu oxide, which indicates that the barrier layer is necessary to inhibit the formation of Cu oxide. In the case of Ni/Ag metallization, a very thin layer of oxide was also formed between Ni and Ag, as shown in Figure 5f. Although Ni is a rather common candidate for the barrier layer in electronic packaging, its usage in WBG devices should be considered with care. Conversely, Ti/Ag and Ni/Pd/Pt/Ag metallization showed excellent thermal stability in HTS tests, as shown in Figure 5g,h. The formation of an oxide layer was prevented by the use of Ti and Pt barrier layers. However, voids appeared in the Ti/Ag metallization after HTS test for 500 h. The formation of voids in the Ag metallization was attributed to Ag atoms’ diffusion from metallization to the Ag porous network [21]. In addition, micro voids were observed in the bonding joint with Ni/Pd/Pt/Ag metallization, which was caused by excessive Ag grain growth and the related coarsening of porous structure.

Figure 6 shows the microstructure evolution of SiC_p_-doped Ag sintered joints at the substrates’ side with various metallization in as-sintered state and after HTS for 500 h. In the case of SiC_p_-doped Ag sintered joints, the interfacial degradations such as void (Figure 5g) and micro crack (Figure 5h) were not observed in Ti/Ag and Ni/Pd/Pt/Ag metallization, because grain growth and coarsening were inhibited by the use of SiC_p_ in sintered joints, as shown in Figure 6g,h.

Figure 7 shows the evolution of the on-resistance of pure Ag joints and SiC_p_-doped joints on various substrates. At 100 h, a < 5% change of on-resistance was observed in pure Ag sintered joints, with various substrates as shown in Figure 7a. With increasing test time, the change of on-resistance with various substrates showed a similar increasing tendency, due to the interfacial degradations between sintered Ag and the substrates. The bonding structures involving DBC substrates and metallization, and their evolution during HTS, were inferred to be the cause of on-resistance change. 

The total variation of on-resistance on Ni/Pd/Pt/Ag metallization was less than 10%, which is obviously superior to that on Ti/Ag and Ni/Ag metallization. The lowest on-resistance change of 0.9% after HTS tests for 500 h was obtained in Ni/Pd/Pt/Ag metallization, and this value was approximately 2% lower than that of SiC_p_-doped Ag sintered joints, as shown in Figure 7b. The relatively lower on-resistance change of Ni/Pd/Pt/Ag metallization corresponded well with the stable interfacial structure during HTS.

In contrast, the on-resistance changes of Ti/Ag and Ni/Ag metallization, in both pure Ag joints and SiC_p_-added joints, were higher than 15%. Ti is known as a material of high electrical resistance, and it has been reported that the Ti/Ag interface has a higher rate of interfacial dewetting [22]. This can be used to explain the increase of on-resistance in Ti/Ag joints. In the case of Ni/Ag, only a thin layer of oxide could be observed after HTS, therefore it is rational to conclude that the thin oxide layer drastically decreased interfacial conductivity.

However, it is worth noting that the thick oxidation layer between Cu and Ag in Ag metallization seemed to have little influence on the changes in on-resistance compared to the thin oxide in Ni/Ag metallization, as shown in Figure 7b. Since copper oxide is a classic semiconductor, it is rational to infer that the oxide formed between Cu and Ag was not a copper oxide, such as Cu_2_O or CuO, and it should have some unrevealed properties, which will be elucidated in our future research.

Figure 8a,b show the evolution of the shear strength with pure Ag joints and SiC_p_-doped joints, respectively, after up to 500 TC tests. The as-sintered shear strength of joints on Ag, Ti/Ag, and Ni/Pd/Pt/Ag metallization were obviously higher than that of Ni/Ag metallization. Since Ni/Ag metallization had thin Ag metallization compared to others, the Ag metallization on SiC side showed a lower diffusion ability during sintering, which indicates the necessity of sputtering an additional Ag layer on the chip side. The shear strength of both joints decreased during TC tests and showed no obvious differences on various metallization layers. As shown in Figure 9, severe coarsening of pure Ag sintered joints occurred during TC tests, which led to cracking at the interface between SiC chips and DBC substrates. In addition, delamination of SiC chips was observed in SiC_p_-doped joints during TC tests, as shown in Figure 10. The delamination of SiC chips was attributed to thermomechanical stress caused by larger coefficient of thermal expansion CTE) mismatch between SiC and Ag. Although coarsening of the Ag network did not occur due to the presence of SiC_p_ inside the joints, cracks were also observed in die-attach structure similar to pure Ag joints, resulting in a decrease of the shear strength (Figure 8). From these results, it was revealed that the tendency of the shear strength during TC tests was entirely different from our previous work, which used same pastes for a Si-DBC die-attachment structure [17]. In our previous work, SiC_p_-doped Ag paste showed excellent thermostability in die-attachment structures involving Si chips and DBC substrates [17].

To clarify the cause of severe cracks in the die-attach structures involving SiC chips and DBC substrates during TC tests, thermal–mechanical simulation was analyzed by using commercial finite element analysis software (ANSYS workbench, Release 15.0, ANSYS Inc, Canonsburg, PA, USA) during the change of temperature from −50 °C to 250 °C, as shown in Figure 11. The stress distribution in the present structure was compared to that in the previous Si-DBC structure [14]. The bonding structure in the previous study consisted of a Si chip and DBC which had thinner Cu and thicker Si_3_N_4_ compared to the present DBC structure. Figure 11a,b show the finite element model of the previous and present die-attachment structures, respectively. Figure 11c shows the stress distribution in the die-attach structure where thermomechanical stress accumulated on the sintered joint. The maximum stress of 35 MPa was obtained due to a CTE mismatch between Si and Cu, which was two times lower than that in present structure (Figure 11d). The reason for the difference in thermomechanical stress was attributed to the mechanical properties of SiC and the design of the DBC substrate. The SiC chip had a larger CTE mismatch with Cu and a higher elastic modulus compared to the Si chip, which indicated that higher thermomechanical stress could be generated in the die-attachment structure. In addition, since thick Cu with thin Si_3_N_4_ present in the DBC substrate is easily deformed, due to lack of stiffness compared to the previous DBC that used thick Si_3_N_4_ and thin Cu, higher thermomechanical stress occurred in the present structure because of the deformation of the DBC during TC tests. Obvious degradation of shear strength was observed during TC testing on SiC-DBC joints, which shows different thermomechanical behavior with Si-DBC joints. These results indicated that the material properties and dimensions should be taken into consideration during the design of a SiC-DBC bonding structure.

## 4. Conclusions

We have studied a thermostable die-attach structure which includes stable sintered porous Ag. This bonding structure was fabricated on common DBC substrates with multi-layer surface metallization. The use of SiC_p_ in the bonding structures inhibited the grain growth and coarsening of the porous Ag network during HTS tests. In addition, Ti and Pt diffusion barrier layers played a key role in preventing interfacial degradations between sintered Ag and Cu at high temperatures. The bonding structure demonstrated excellent high temperature stability, maintaining a stable structure after HTS tests. The changes of on-resistance indicated that Pt metallization has relatively better electrical properties compared to that of Ti and Ni. The thick oxide layer formed on an Ag/Cu surface had different electrical behavior compared to that of normal copper oxide, which needs further research. In addition, the stress distribution in the present structure was compared to that in a previous Si-DBC structure. The higher thermomechanical stress was generated in the present die-attachment structure due to a larger CTE mismatch between the SiC chip and Cu. The different thermomechanical behaviors of SiC-DBC and Si-DBC structures indicated that the material properties of Si and SiC should be taken into consideration during the design of a SiC-DBC bonding structure.

## Figures and Tables

**Figure 1 materials-11-02531-f001:**
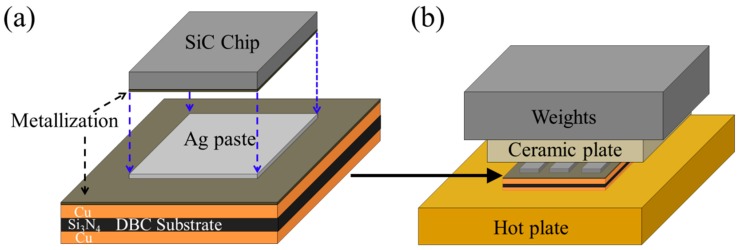
Diagrams of (**a**) SiC- direct bonded copper (DBC) joint, and (**b**) pressure-aided sintering.

**Figure 2 materials-11-02531-f002:**
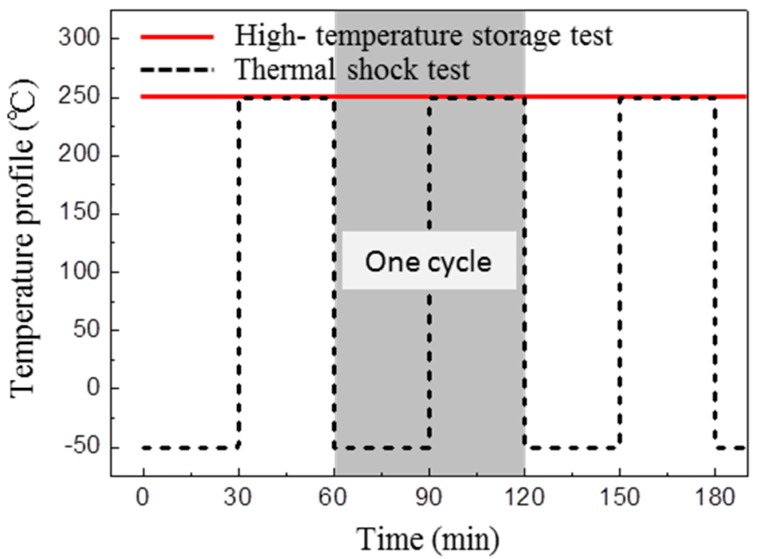
Profiles of harsh environmental tests: high temperature storage test at 250 °C and thermal cycling test at −50/250 °C.

**Figure 3 materials-11-02531-f003:**
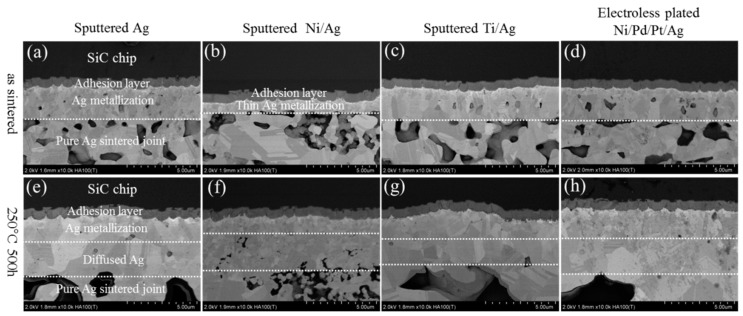
Evolution of pure Ag sintered joint at the chip sides in as-sintered state and after high temperature storage (HTS) for 500 h: (**a**–**d**) as sintered state, (**e**–**h**) after HST for 500 h.

**Figure 4 materials-11-02531-f004:**
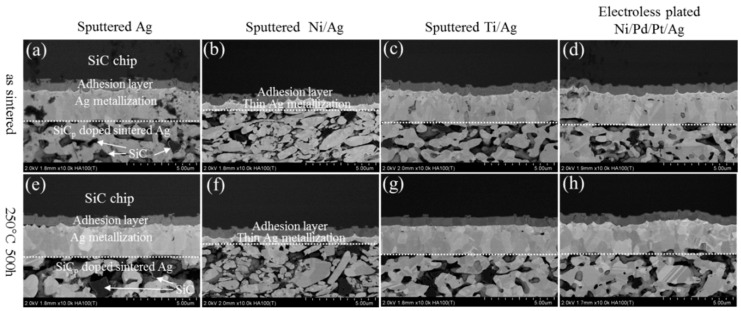
Evolution of SiC_p_ doped Ag sintered joint at the chip sides in as-sintered state and after HTS for 500 h: (**a**–**d**) as sintered state, (**e**–**h**) after HST for 500 h.

**Figure 5 materials-11-02531-f005:**
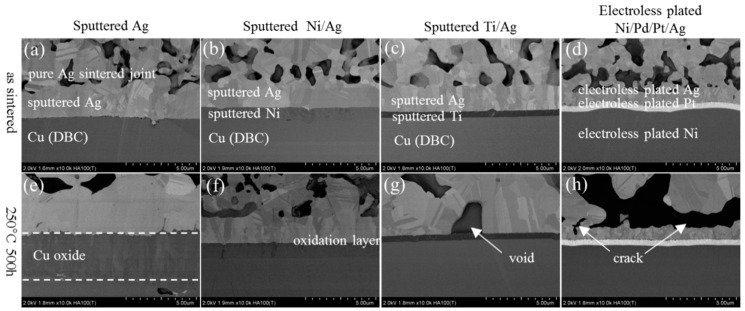
Evolution of a pure Ag sintered joint with various substrates in as-sintered state and after HTS for 500 h: (**a**–**d**) as sintered state, (**e**–**h**) after HST for 500 h.

**Figure 6 materials-11-02531-f006:**
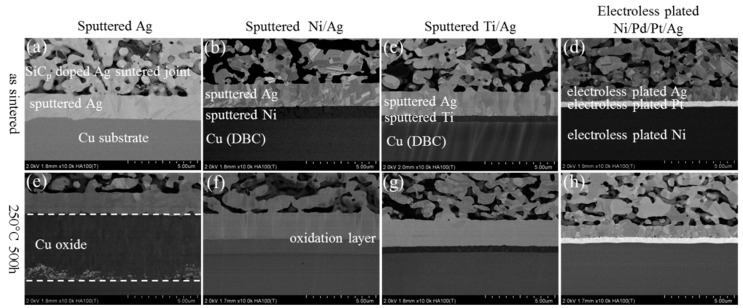
Evolution of SiC_p_ doped Ag sintered joint with various substrates in as-sintered state and after HTS for 500 h: (**a**–**d**) as sintered state, (**e**–**h**) after HST for 500 h.

**Figure 7 materials-11-02531-f007:**
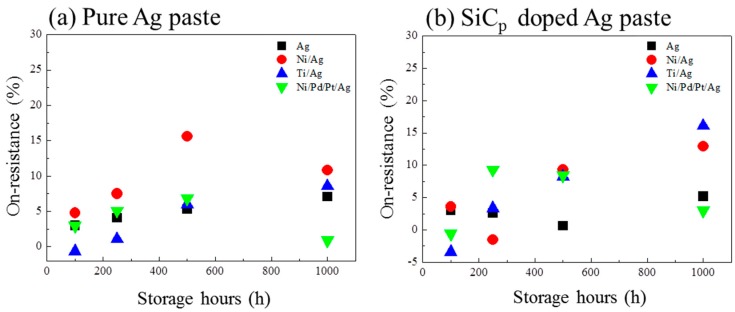
Evolution of the on-resistance of (**a**) pure Ag paste and (**b**) SiC_p_-doped Ag paste on various substrates.

**Figure 8 materials-11-02531-f008:**
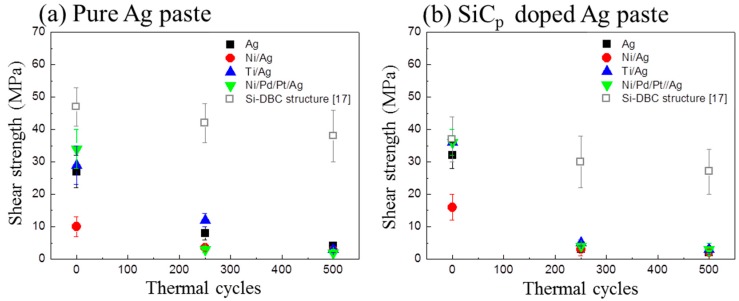
Evolution of the die shear strength of (**a**) pure Ag paste and (**b**) SiC_p_-doped Ag paste on various substrates, the former die-shear results of Si-DBC joints are listed as reference [17].

**Figure 9 materials-11-02531-f009:**
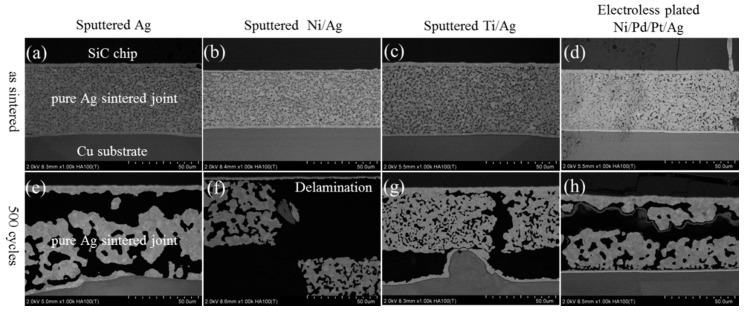
Evolution of pure Ag sintered joint with various substrates in as-sintered state and after a TC test of 500 cycles: (**a**–**d**) as sintered state, (**e**–**h**) after 500 cycles.

**Figure 10 materials-11-02531-f010:**
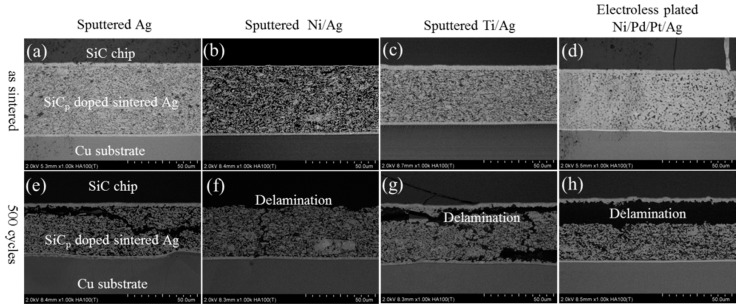
Evolution of SiC_p_ doped Ag sintered joint with various substrates in as-sintered state and after TC for cycles: (**a**–**d**) as sintered state, (**e**–**h**) after 500 cycles.

**Figure 11 materials-11-02531-f011:**
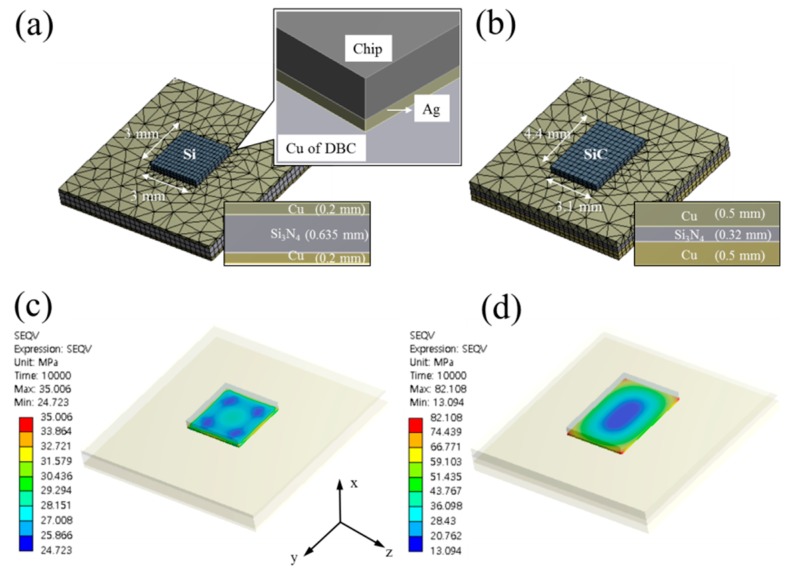
The model of die-attachment structures for finite element simulation: (**a**) previous work [17], (**b**) present work, the stress distribution of model under the change of temperature from 25 °C to 250 °C: (**c**) previous work, (**d**) present work.

**Table 1 materials-11-02531-t001:** Properties of Ag particle and SiC_p_.

	Average Diameter (μm)	Thickness (nm)	Specific Surface Area (m^2^/g)
Ag flakes	6.0	80	5.0
SiC_p_	0.6	N/A	N/A

**Table 2 materials-11-02531-t002:** Specifications of direct bonded copper (DBC) substrates and SiC chips.

SiC Metallization Scheme	DBC Metallization Scheme
Ti/Ni/Au/Ag: Ag thickness 2000 nm	Ag: 2000 nm
Ti/Ni/Au/Ag: Ag thickness 300 nm	Ni/Ag: 2000 nm/2000 nm
Ti/Ni/Au/Ag: Ag thickness 2000 nm	Ti/Ag: 500nm/2000 nm
Ti/Ni/Au/Ag: Ag thickness 2000 nm	Ni/Pd/Pt/Ag: 5000 nm/30 nm/300 nm/2000 nm

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
