# Peer review of "Heat-Resistant Microporous Ag Die-Attach Structure for Wide Band-Gap Power Semiconductors"

_materials, 2018, doi:10.3390/ma11122531_

Round 1
Reviewer 1 Report
Thank you for the manuscript. My comments are below.
Major comments:
The authors claim the delaimation of SiC chips were observed in SiCp doped joints during TC tests (line 202-203). The authors also claim SiCp doped Ag paste showed excellent thermostability (line 208-209). Please explain how these two statements are consistent.
A more informative explanation is needed for the inhibition of coarsening of Ag porous structure when SiCp doped Ag is used (lines 132-133). As of now, there is no mention regarding the Ag diffusion when SiCp doping is used. Also, “hard nature” is not a clear term, and “high temperature stability” is not answer to the prevention of grain growth.
The oxide formation arguments given in line 187-191 are not clear. How is it rational to assume the formed oxide is not Cu2O or CuO? What do you mean by “unrevealed properties”?
Please explain how did you come up with the thickness values of the metallization layers shown in Table 2.
The authors state that they used 5 specimens for the shear strength test as indicated in Fig. 8 where they provided error bars. However, there are no error bars in Fig. 7. If possible, the authors should provide error bars for the resistance changes in Fig. 7.
The delamination mentioned in the paragraph between the lines 194-209 should be clearly shown in Fig. 10. Where do you we see the delamination? Also, provide plausible reasons for the delamination phenomenon.
It is not clear how a comparison of die shear strength between the authors’ previous study and current study (line 206-208). Provide reasons.
It is not clear what the authors mean while stating the use of Ni in WBG devices requires care. A concise argument should be provided here.
It is not clear what the authors mean by “various metallization structures” in line 171-172. A clearer explanation is needed.
The micro crack mentioned in line 159 should be explicitly shown in Fig. 5h. Where is the crack?
The authors should explain their previous structure briefly before talking about it (line 219-220).
Minor comments:
In line 17, give the extension of DBC as “direct bonded copper” before you use the abbreviation.
In abstract, the statement starting with “While Ni barrier...” is not a sentence per se. It should be combined with the previous sentence.
In line 31, Give the band-gap values values of GaN and SiC to educate the reader.
In lines 34-37, be explicit about the current interconnection materials. What are those materials? What are their specific property that prevents them to be used continuously above 150 deg C?
In line 52, “Ag micron flake” should be corrected as “Ag micron-sized flake”.
As of now, the introduction section ends as an experimental section. End the introduction section with the most important finding in this study.
In line 71, remove “dissolved” since the mentioned physical process carried out in ethanol is not dissolution in the conventional sense, but merely a suspension formation.
In line 81, use the extensions of HTS and TC along with their abbreviations.
In line 80, fix “3.1x4.4x0.35 mm^2” as “3.1x4.4x0.35 mm^3”.
In line 87, fix “as-received ones” as “as-received”. Also, add verb “used” before “as-received”.
In line 89, it is not clear how the Ag paste was coated onto the substrate. The coating process should be explained briefly.
In line 94, replace “attached to” with “bonded on”.
In lien 98, remove the sentence starting with “Table 2 present...” since Table 2 is already mentioned in line 89. As of now, the sentence is just a repetition.
Fix the Table 2 such that the last row is on the same line. As of now, the last row is confusing.
In line 180 the second sentence, fix “are” as “is”.
In line 181, the authors should explain why Ti/Ag interface has higher rate of interfacial dewetting. Also, the authors should cite literature sources for that claim.
Throughout the manuscript, fix “researches” as “research” or “studies”.
Author Response
Dear reviewer
The authors thank you very much for giving us an opportunity to revise our manuscript and we appreciate you very much for your positive and constructive comments and suggestions concerning our manuscript entitled "Heat-resistant microporous Ag die-attach structure for wide band-gap power semiconductors ". Those comments are all valuable and very helpful for revising and improving our paper, as well as the important guiding significance to our researches. We have studied comments carefully and have made corrections which we hope meet with approval. Revised portions are marked in red in the paper.

Reviewer 2 Report
Authors must explain in details why they choose this specific thermal cycling temperature (-50 to .250oC
In figure 7 (a) after storage test for 1000hr, the on-resistance value was decreased as compared to the sample storage for 600hr, please explain in details the reasons.
In shear strength graph, the strength value was dropped significantly, if possible please measure the shear strength after exposure to 100 hr.
Authors must rewrite the conclusion according to their experimental results.
Author Response
reviewer
The authors thank you very much for giving us an opportunity to revise our manuscript and we appreciate you very much for your positive and constructive comments and suggestions concerning our manuscript entitled "Heat-resistant microporous Ag die-attach structure for wide band-gap power semiconductors "Those comments are all valuable and very helpful for revising and improving our paper, as well as the important guiding significance to our researches. We have studied comments carefully and have made corrections which we hope meet with approval. Revised portions are marked in red in the paper.
